# Semi-Autonomic AI LF-NMR Sensor for Industrial Prediction of Edible Oil Oxidation Status

**DOI:** 10.3390/s23042125

**Published:** 2023-02-13

**Authors:** Tatiana Osheter, Salvatore Campisi Pinto, Cristian Randieri, Andrea Perrotta, Charles Linder, Zeev Weisman

**Affiliations:** 1Phyto-Lipid Biotech Lab (PLBL), Department of Biotechnology Engineering, Ben Gurion University of the Negev, Beer Sheva 8499000, Israel; 2eCampus University, Via Isimbardi, 10, 22060 Novedrate, Italy

**Keywords:** T_2_ relaxation curve, self-diffusion, chemical standards of oil oxidation, data mining, CNN, machine learning, pattern recognition

## Abstract

The evaluation of an oil’s oxidation status during industrial production is highly important for monitoring the oil’s purity and nutritional value during production, transportation, storage, and cooking. The oil and food industry is seeking a real-time, non-destructive, rapid, robust, and low-cost sensor for nutritional oil’s material characterization. Towards this goal, a ^1^H LF-NMR relaxation sensor application based on the chemical and structural profiling of non-oxidized and oxidized oils was developed. This study dealt with a relatively large-scale oil oxidation database, which included crude data of a ^1^H LF-NMR relaxation curve, and its reconstruction into T_1_ and T_2_ spectral fingerprints, self-diffusion coefficient D, and conventional standard chemical test results. This study used a convolutional neural network (CNN) that was trained to classify T_2_ relaxation curves into three ordinal classes representing three different oil oxidation levels (non-oxidized, partial oxidation, and high level of oxidation). Supervised learning was used on the T_2_ signals paired with the ground-truth labels of oxidation values as per conventional chemical lab oxidation tests. The test data results (not used for training) show a high classification accuracy (95%). The proposed AI method integrates a large training set, an LF-NMR sensor, and a machine learning program that meets the requirements of the oil and food industry and can be further developed for other applications.

## 1. Introduction

### 1.1. Importance of Evaluation of Oxidation of Edible Oils

The oxidation stability of edible oils and lipid-rich foods is a very important issue, especially with foods that contain polyunsaturated fatty acids (PUFAs), an important nutritional component for health such as cardiovascular and neurological health [1,2,3]. Paradoxically, however, these lipids are highly susceptible to oxidation into toxic products during food preparation, transportation, storage, and cooking [4,5,6], affecting the food’s texture, taste, aroma, and shelf life [7].

Controlling foods’ PUFA oxidation is difficult as even endogenous or added antioxidants are not sufficiently effective without optimal structural arrangement of the food components that change during processing, storage, and digestion [4,8]. Thus, monitoring a food’s oxidation status in real time during its preparation can contribute significantly to the product’s quality by guiding the choice of the optimal food formulations.

The oil and food industry is seeking significant advances in the evaluation of oil oxidation that will meet the following requirements: non-destructive, real-time, rapid testing; reliable; and robust methodologies that do not require elaborate calibrations nor the use of hazardous chemicals and are as low cost as possible [7,9].

### 1.2. Oxidation Process of Edible Oils

Oxidation is characterized by initiation, propagation, and termination phases, where the initiation step may be triggered by relatively high temperatures, incident light, transition metals, or initiators [10,11]. Initiation is formed on alkyl radicals (R^·^) and in PUFA on allylic carbons by hydrogen atom removal, followed by double-bond isomerization into a conjugated structure. Hydrogen removal kinetics is dependent on its molecular position such as the bisallylic hydrogen atom having the highest dissociation rate. Lipid alkyl radicals (R^·^) may react with atmospheric oxygen and form peroxy radicals (ROO), which propagates the reaction via hydrogen atom abstraction from another lipid molecule forming hydroperoxides (ROOH) and another lipid alkyl radical (R^·^). A major termination step is the chain-reaction radicals reacting with each other into covalent bonds [11,12,13].

Hydroperoxides, known as the primary oxidation products, usually undergo further oxidation to give secondary oxidation products. High temperatures or transition metals enhance hydroperoxide decomposition into alkoxy radicals that subsequently form aldehydes, ketones, acids, esters, alcohols, short-chain hydrocarbons, and higher molecular weight oligomeric or polymeric molecules [14,15]. Secondary oxidation results in the food’s off-flavor as well as being associated with pathological sclerosis, the degradation of biologically active proteins, and the aging process [16].

### 1.3. Available Technologies for Evaluation of Oil Oxidation

Numerous chemical and physical analytical methods have been developed to assess lipid oxidation such as conjugated diene values, peroxide values (PV), alcohols, epoxides, *p*-anisidine value (*p*-AV) assay, HBR titration, iodometric titration, xylenol orange, and total polar components (TPC) by analytical systems, e.g., HPLC, GC-MS, FTIR, SEC, ESR, and high-field (HF) NMR [6,17,18].

Many of these technologies are based on methods with chemical reactions requiring relatively long reaction times and extensive labor, which cannot be automated; measure only one kind of oxidation product; only determine the concentration of an oxidation product that reaches a peak in a short time; and/or have inconsistencies due to multiple variations in the procedure [6]. Overcoming current analytical limitations requires analytical instruments for measuring multiple samples in a short time and detecting many oxidation products in a given sample during the oxidation processes. The proton LF-NMR systems can potentially meet these needs for the analysis of lipid components in foods. More conveniently, user-friendly NMR instruments such as small benchtop LF-NMR instruments and software programs such as an automated analysis program for the fatty acid composition of vegetable oils have been developed [6].

Low-field NMR relaxation technology is recognized to have significant potential in elucidating the molecular structures of lipid oxidation products and the oxidation mechanisms. It was recently shown that LF-NMR relaxation is highly efficient in rapidly and accurately monitoring oil oxidation [4,5,19], as briefly described below.

### 1.4. ^1^H Low-Field NMR Relaxation Sensor Technology

The low-field NMR relaxation sensor is a rapid, versatile, non-destructive analytical tool extensively used in the food, polymer, petroleum, and pharmaceutical industries [20,21,22,23], and is increasingly attracting the food industries’ attention for quality control without the need for chemicals with health or environmental effects [6,24].

This NMR relaxometry tool is useful for identifying molecular species and their dynamics even in complex materials by the measurement of energy relaxation time constants as a consequence of interactions between nuclear spins and between spins and the surrounding lattice. The time constants for longitudinal (spin–lattice) and transverse (spin–spin) energy relaxations are T_1_ and T_2_, respectively.

Relaxation time distribution experiments range from rapid one-dimensional (1D) tests to more complicated multidimensional ones. The most commonly applied, 1D NMR tools, are based on either the acquisition of the free induction decay signal following a 90° pulse or pulse sequences such as the spin echo [25], CPMG [26,27], or inversion/saturation recovery [28].

The basic principle of the LF-NMR sensor is that the oil FA composition directly affects and dominates the proton energy relaxation time, as shown by three different oils in Figure 1. The most viscous castor oil, composed of about 90% ricinoleic acid (12-OH-18:1) with hydroxyl groups, has T_2_ aliphatic chain proton energy relaxation time values significantly shorter than the other two oils. Monounsaturated olive oil, of lower viscosity, with about 60% oleic acid (18:1) shows an intermediate T_2_ relaxation time, while PUFA-rich linseed oil, of lowest viscosity, with about 55% linolenic acid (18:3) is the most mobile and the T_2_ relaxation time is the longest of the three oils. This clearly shows the effect of both the chemical composition and its structural arrangement on the NMR signal generation in terms of proton time spin–spin (T_2_) energy relaxation values. However, significantly more information on the oils’ chemical and morphological material arrangements can be obtained from this data by more advanced algorithms for signal collection and T_1_ and T_2_ spectrum reconstruction. Related time spectra are generated by processing ^1^H LF-NMR signals as ‘raw data’ using a modified primal-dual interior method for convex objectives (PDCO) optimization solver that optimized L_1_/L_2_ norm regularization parameters for inverse Laplace transformation (ILT) to rapidly generate the 2D T_1_–T_2_ fingerprint spectra of organic materials [29]; the different spectrum peaks were then assigned their chemistry and morphologies based on the rigidity/mobility of their proton populations [29,30,31,32]. 

By applying the LF-NMR to an oil, of a fairly complex structure, four different rigidity–mobility segments of the alkyl chain are detected [33] where the most physically rigid microstructure is the glycerol segment (Figure 2) and secondly the alkyl chain’s double-bond segment. The more mobile segments are the saturated aliphatic chains between the head segment and the double bonds with the aliphatic tail chain, from the outer most double bond, being the most mobile segment [33,34,35].

Food oils based on triacylglycerides (TAGs) are readily characterized by proton time-domain (TD) NMR and T_1_ and T_2_ reconstruction into TD spectra. Figure 3A is linolenic-rich linseed oil’s (LSO) 2D T_1_–T_2_ relaxation time spectral fingerprinting. LSO’s chemical structure is characterized by four different segmental motion TDs (Figure 2). The NMR relaxation analyses give a spectrum with four peaks that are correlated with the PUFA-rich LSO segmental motion TDs.

The analysis of the molecular configuration and aggregate structures of PUFA oils with respect to their susceptibility and degree of oxidation is important for foods because of PUFA-rich oil’s facile oxidation into toxic side products [19,36,37]. The oxidation process generates volatile, low-molecular-weight, and polymeric structures. The magnitude of the relaxation times T_1_ and T_2_ are dependent upon the interaction between ^1^H spins and their surrounding environment (lattice) that is modulated by molecular mobility. This phenomenon is related to the chemical and structural changes used to develop the LF-NMR relaxation sensor application to monitor a lipid’s oxidation.

Linseed oil (LSO) oxidation results in the oil’s chemical and physical structural changes, as followed by an LF-NMR sensor with high efficacy [19]. After 120 h of thermal oxidation at 80 °C, the 2D T_1_-T_2_ energy relaxation time spectra show new peak formation of oxidation products, resulting from crosslinking and polymerization in the termination phase of oxidation. The relatively high molecular weight of the oxidation products is characterized by different values of T_1_ and T_2_ per peak, forming the characteristic oxidized bending effect from the T_1_ = T_2_ diagonal in the spectra curve. The four characteristic LSO TDs (peaks 1–4) remain, but they have lower values because of the secondary interactions between the non-oxidized LSO and its oxidation products.

The numerical values of T_1_-T_2_ of the peaks’ pattern/signature shown in Figure 3A are suitable to be used for the pattern recognition of non-oxidized linseed oil and the numerical values of T_1_-T_2_ peaks shown in Figure 3B for the pattern recognition of highly oxidized and toxic linseed oil [4].

An important material characterization is the self-diffusion coefficient, D, that is determined by the ^1^H LF-NMR pulsed-field gradient spin echo (PFGSE) method [38], which is well accepted for characterizing ^1^H mobility based on the chemical and physical status of edible oil with fatty acids and esters [39,40]. Using D values of intact LSO and thermal oxidized samples, the pattern of mobility changes occurring during LSO oxidation could be monitored with a very good correlation between coefficient D and conventional wet chemistry standard tests of oil oxidation based on PV, *p*-AV, and TOTOX [41]. These results increased the confidence in using LF-NMR relaxation technology for a rapid real-time and on-line evaluation of edible oil oxidation status as presented in this paper. It should be also noted that this intelligent LF-NMR relaxation sensor application was successfully used beyond linseed oil for many different oil types and additional, more complex lipid-rich food products sensitive to oxidation. The basic ability described above to carry out a relatively rapid (in minutes scale) oil oxidation profiling and to determine a chemical and structural mobility pattern recognition profile enables the classification of non-oxidized and highly oxidized oil, allowing the development of an even more rapid automation step (in seconds scale) based on a machine learning (ML) model described below.

### 1.5. Machine Learning (ML) Technology

Convolutional neural networks (CNNs) are machine learning algorithms that can be used, among other things, for multinomial classification needed for rapid real-time analysis. That is the problem of classifying instances (i.e., sets of observations) into three or more different classes. Classification is a form of “pattern recognition”. The process of adapting an off-the-shelf model (e.g., a CNN) to a particular problem (in this case: classification of T_2_ signals) is called “fine tuning” and “machine learning” or simply “learning”. Once the learning phase is concluded, the model can be deployed. In this case, the trained CNN will infer the oxidation class of the oil that generated the T_2_ signals.

When large numbers of labeled instances are available, machine learning is an attractive solution for classification tasks because it does not require any deterministic function about input values and desired output. A fundamental difference between machine learning and classical statistical models (e.g., linear regression) is their purpose.

Classical statistical models typically deal with the collection, analysis, interpretation, and presentation of data about a system. These are often based on restrictive assumptions about the data as well as the system that generates the data, so that a model is superimposed on the data at hand.

Machine learning, on the other hand, involves algorithms that can learn very complex (often unknown) relationships between the variables without any assumption about the data and/or the system generating the data. Machine learning algorithms can be divided into two main categories: supervised learning algorithms, which learn from labeled training data, and unsupervised learning algorithms, which learn from unlabeled data. Supervised learning was used, where conventional lab tests served as ground-truth labels as they exhibit known oxidation levels. In supervised learning, the model learns by example, so that along with the input variable (T_2_ raw signals) the model was also given the corresponding ground-truth labels representing total oxidation. During training, the model learns a function (i.e., a set of hyper-parameters) that maps out instances (i.e., T_2_ signals) to correct target classes (i.e., oxidation levels).

Thus, when a large amount of data is available and prediction is a priority over description, machine learning is the proper solution.

A convolutional neural network (CNN) is an important subgroup of machine learning algorithms. It is particularly useful for problems with special and/or time-dependent input features. CNNs gained popularity in 2012 (the AlexNet for image recognition) and have since been used in countless applications. CNNs have been also used for a variety of 1D-signal applications [42], i.e., Google’s Wavenet for text-to-audio generation [43], signal denoising [44,45], large-scale audio classification [46,47], and speech processing [48,49]; data upscaling and spike detection in ATAC-seq sequence data [44,50,51]; quantitative [52,53,54] and qualitative [54] spectroscopy; and chromatographic [55] applications. These applications deal with 1D signals as the one acquired by LF-NMR. Alternatives to CNN, for the same types of data, include decision tree learning, clustering, support vector machines (SVMs), K-means nearest neighbor (K-NN), restricted Boltzmann machines (RBMs), and random forests; the prerequisite for such techniques to work efficiently is the extraction of discriminating features. That is a time-consuming and often inefficient task, especially for this type of input signal where features are not interpretable by means of direct human observation. On the other hand, a convolutional neural network (CNN) can automatically learn the necessary features and extracts them for further classification. Thus, CNNs combine automatic feature extraction with supervised learning (e.g., artificial neural networks). Currently, this is one of the most powerful classification tools for both scientific and commercial applications.

The main objective of the present study was to develop a portable ^1^H LF-NMR relaxation semi-autonomic sensor for real-time evaluation of edible oil oxidation to meet the call and requirements of the oil and food industry to produce optimal food products without lipid oil’s oxidation. The specific objectives were to generate a large-scale database including LF-NMR signal acquisitions as well as conventional lab measurements of oil samples at different oxidation levels to be used for data mining and classification; to develop a suitable machine learning model for the classification of edible oil oxidation level (non-oxidized oil, partially oxidized oil, highly oxidized oil); and to validate and test the ML classification accuracy.

## 2. Materials and Methods

### 2.1. Materials

All chemicals and reagents used in this study were analytical grade. Freshly extracted oil labeled as linseed oil (*Linum usitatissimum* L.) was purchased just before the beginning of the experiments from a local supplier (Nes Shemanim, Israel) and stored within a tightly closed glass container in a cool shaded environment. The fatty acid profile of the oil was tested by GC-MS just before the beginning of the analyses used in the present study and was confirmed to be standard linseed oil.

### 2.2. Thermal Oxidation

Thermal oxidation was carried out on linseed oil. Autoxidation experimental design was based on previous studies [33,56] wherein autoxidation was induced by heating 150 mL of the sample in a 250 mL beaker on an 80 °C hot plate. Air was pumped into the beaker with maximum stirring, for 120 h, using a glass Pasteur pipette and a vacuum pump (Vacuubrand MZ 2C Diaphragm Vacuum Pump, Isleham, UK). At the start and after 24, 48, 72, 96, and 120 h of the oxidation process, a 10 mL sample was removed for analysis. The analysis included commonly used chemical analysis measurements of peroxide values (PV), aldehyde values (*p*-Anisidine), and calculation of total oxidation values (TOTOX) described below.

### 2.3. Conventional Standard Wet Chemistry Analyses

The primary oxidation products were evaluated with peroxide value (PV) tests according to the AOAC Official Method 965.33.12 (Official methods of analysis of AOAC international, 17th ed. Rockville, MD, USA). The *p*-anisidine (*p*-AV) test was used in the assessment of secondary oxidation products according to the AOCS Official Method Cd 18–90 (2002) [16,57].

Total oxidation value (TOTOX) was calculated by the formula *p*-AV + 2PV to indicate an oil’s overall oxidation state. The lower the TOTOX value, the better the quality of oil [58].

### 2.4. Low-Field NMR Relaxation Analysis

The spin–spin ^1^H LF-NMR measurements were carried out with a Maran benchtop pulsed NMR analyzer (Resonance Instruments, Witney, UK) with a permanent magnet and an 18 mm probe head operating at 23.4 MHz. Before each measurement, the samples were stabilized at 40 °C for 40 min and then allowed to equilibrate inside the instrument for 5 min. The spin–spin relaxation time constant (T_2_) was generated using a Carr–Purcell–Meiboom–Gill (CPMG) pulse sequence. The CPMG sequence consists in applying a 90-degree radiofrequency pulse to the sample, followed by many 180-degree pulses. Each time a 180-degree pulse was applied, the signal decay of the magnetic field was removed and a single data point was acquired [26,27].

For every sample, several scans were accumulated. Initial batches had only one scan per sample, and later batches had 35 scans. The number of echoes acquired was 16,384 with a recycle delay of 6 s and *τ* between 200 and 550 μs. Receiver gain (RG) and magnetic field were calibrated before each measurement. The signal processing was based on the PDCO inverse Laplace transform optimization algorithm with *a*_2_ = 0.5 [29,31].

### 2.5. Self-Diffusion

The self-diffusion measurements were carried out with a 20 MHz minispec benchtop pulsed NMR analyzer (Bruker Analytic GmbH, Berlin, Germany), equipped with a permanent magnet, and a 10 mm temperature-controlled probe head according to [56]. Prior to each measurement, the samples were stabilized for 40 min at 40 °C and then equilibrated inside the instrument for 5 min.

The self-diffusion coefficient, D, was determined by the pulsed-field gradient spin echo (PFGSE) method [38]. The pulse sequence was used with 16 scans, *τ* of 7.5 ms, and a recycle delay of 6 s. Typical gradient parameters were *Δ* of 7.5 ms, *δ* of 0.5 ms, time between the 90° pulse to the first gradient pulse of 1 ms, and *G* of 1.6 T/m. Each reported value of the self-diffusion coefficient (D) is the average of 10 measurements.

### 2.6. Machine Learning Methodology

#### 2.6.1. Nuclear Magnetic Resonance Acquisition, Labeling, and CNN Training

In this study, CNN training is part of a larger infrastructure that includes: oil sample treatment (to induce oxidation), T_2_ signal acquisition (via NMR), and CNN training and testing. In particular, the system was organized as follows: each oil sample (Figure 4, step (1)) was treated with one of six different thermal treatments to induce different levels of oxidation on food-grade, pure, high-quality linseed oil (LSO). Treatment duration varied, from a minimum 0 h (control, no induced oxidation) to a maximum 120 h treatment where oxidation is expected to be at the highest levels (treatments denoted as: 0 h, 12 h, 24 h, 48 h, 96 h, 120 h). Subsequently (Figure 4, step 2), each sample was analyzed with both: (i) LF-NMR, for the acquisition of raw T_2_ relaxation curves and (ii) LF-NMR gradient pulse analysis of self-diffusion coefficient D and conventional standard chemical lab methods (PV, *p*-AV, and TOTOX were calculated). Oxidation measurements were converted into ordinal classes as follows: ‘Good’—non-oxidized oil, ‘Fair’—partially oxidized oil, and ‘Bad’—highly oxidized oil, according to the classification criteria presented in Section 2.6.3. These are the ground-truth for the oxidation levels achieved with the treatments in the previous step. The LF-NMR raw T_2_ relaxation signal acquisitions were not transformed into the frequency domain, allowing fast collected raw signals to be used as such. By binding together these measurements (Figure 4, step 3), a basic database of labeled T_2_ signals was formed. Subsequently, in Figure 4, step 4, several convolutional neural networks (CNNs) were trained, benchmarked, and fine-tuned for the classification task of T_2_ signal into three classes corresponding to three different oxidation levels (‘Bad’, ‘Fair’, and ‘Good’).

The CNN consists of (i) the encoding modules, a series of convolutional as well as pulling operations that are used for automatic feature extraction and data dimensionality reduction, followed by (ii) the decoding module, a supervised network of liner combiners (as well as the corresponding activation functions) that was trained by a gradient descent algorithm (i.e., ADAM, the adaptive moment estimation method [59]). The convolutional layers of a CNN apply a set of filters to the input data to learn local features, such as edges and shapes. The pooling layers reduce the dimensionality of the data by summarizing the output of the convolutional layers over a local region. The output of the CNN is then passed through one or more fully connected layers, which apply weights to the output of the convolutional layers to make a prediction or decision. Finally, deep convolution neural network (DCNN) outputs (Figure 4, step 5) are one of the three possible oxidation classes. Different CNN configurations were benchmarked for prediction accuracy. This result focuses only on the final optimized configuration.

#### 2.6.2. The Linseed Oil Oxidation Analyses Database

In order to construct a reliable machine learning CNN model to assess the oxidative status of LSO, a database was created. This database includes 3300 files of CPMG raw relaxation curves and post-PDCO T_2_ spectra of each. These files were created from LSO at different stages of oxidation from 12 different repetitions of the LSO oxidation experiment (also referenced as ‘batches’). Despite every repetition being conducted with the same protocol, the rate of oxidation and thus the final oxidative state of every oil sample was slightly different. This may be explained by other factors that cannot be controlled in the experiment. For this reason, the time at which samples were taken was not a factor taken into account for deciding if oil was oxidized or not. This variance is beneficial when the resulting data are applied in the ML model, as it shows different possible spectra that can be correlated to every category of oil.

In addition to the data files, the database contains results taken from PV, *p*-AV, TOTOX, and D experiments. For every file corresponding to an oil sample, the mentioned values are described. Using these parameters, it is possible to assess whether each oil sample is under the ‘Good’—non-oxidized oil, ‘Fair’—partially oxidized oil, or ‘Bad’—highly oxidized oil category, as further described below.

#### 2.6.3. Criteria for Dividing Oil Samples into ‘Good’, ‘Fair’, and ‘Bad’ Oil Categories of Oil Oxidation

As mentioned above, LF-NMR can be used not only for energy relaxation time measurements but also to determine the self-diffusion (D) of a given sample [41]. When applied to oil oxidation, it is possible to observe the changes that occur at the late stage of this reaction, as secondary products polymerize and change the viscosity of the oil. Previous studies [41] have demonstrated how D can be used to evaluate an edible oil’s oxidative status. For linseed oil, with non-oxidized conditions of measurement, the D value is above 0.030 × 10^−9^ m^2^/s, partially oxidized is in the range of 0.030–0.020 × 10^−9^ m^2^/s, and highly oxidized oil will have a D below 0.020 × 10^−9^ m^2^/s.

In addition to ^1^H LF NMR, numerous chemical and physical analytical lab conventional standard methods have been used to assess lipid oxidation. These include peroxide values (PV) and para-anisidine (*p*-AV) that are used in previous studies [5,41]. These common methodologies have many drawbacks, such as strict time regimes during individual stages of analyses, the need to control reaction conditions and components including light and atmospheric oxygen exposure, and large amounts of environmentally harmful solvents [56]. Though these tests allow the evaluation of either primary or secondary oxidation products, any one of these tests does not give a measure of both products.

By combining the conventional standard chemical methods and self-diffusion coefficient D, it is possible to create a broad profile for LSO samples and their oxidation. Since PV and *p*-AV were found to correlate with coefficient D that correlate to the initial and later stages of oxidation respectively, it is expected that PV values will increase and afterwards the D value will decrease, as previously reported [5]. The PV and D values were used to categorize LSO into 3 groups, described in Table 1. The cutoff value for PV was 30 mmol/kg and for D was 0.03 × 10^−9^ m^2^/s for non-oxidized, ‘Good’ LSO. The cutoff range was 30–50 mmol/kg of PV and a D range of 0.03–0.02 × 10^−9^ m^2^/s for partially oxidized, ‘Fair’ LSO. The cutoff was PV higher than 50 mmol/kg and D values lower than 0.02 × 10^−9^ m^2^/s for highly oxidized, ‘Bad’ LSO. With those criteria, 126 samples belonging to the ‘Good’ category, 77 samples belonging to the ‘Fair’ category, and 187 samples belonging to the ‘Bad’ category were used in the study (Table 1).

### 2.7. Convolutional Neural Network (CNN) Architecture

The convolutional neural network setup (Figure 5) included 4 dilated convolution layers. A ReLU (Rectified Linear Unit) was applied to each of these layers as an activation function g(z) = max{0, z} that connects the convolution layer and the pooling layer; the non-linear feature maps obtained from the convolution layers are passed to the pooling layer. The ReLU is the most widely used activation function with the convolution layer. Numerical experiments confirmed that ReLU reduced vanishing gradients and sped-up convergence. Batch size dimensions were automatically defined by the Tensorflow–Keras library. One-dimensional, normalized (0 to 1) input data were used. Normalization improved training efficiency; it also reduced overfitting resulting from excessively large weights that reflect the original units of the input. After the dilated convolution layers, a max-pooling layer was applied to reduce the size of the output from the previous layer and improve feature detection regardless of the sampling difference (if any) in the T_2_ source inputs. Thus, at least in theory, the model can handle different input shapes, acquired from different types of LF-NMR machines and/or settings.

After the convolutional module, a fully connected classifier was set up. Adam’s gradient descent was used [54] to minimize the loss function. The method is a back-propagation method that is adaptive in such a way that it tunes the learning rates for each parameter and at each epoch, thus maintaining an exponentially decreasing average of the past and squared gradients, using them to update the parameters. The Adam optimization function improves the gradient descent method by reducing the risk of local minimum convergence. The number of training iterations (epochs) was chosen empirically to balance out accuracy versus overfitting. Finally, a “softmax” output layer assigns estimated probabilities over the various classes. The trained model returns the oxidation class (either ‘Good’, ‘Fair’, or ‘Bad’) and the corresponding estimated probability.

## 3. Results and Discussion

### 3.1. Low-Field NMR Relaxation Curves of Thermally Induced LSO Oxidation

As described above in Section 1.3 of the introduction, the LF-NMR sensor relaxation technology developed includes an initial step of the collection of proton T_1_ and T_2_ relaxation signals, forming relaxation curves, followed by an inverse Laplace transformation (ILT) data processing step that results in a graphic fingerprinting spectra. These two steps provide very valuable chemical and structural information but require a relatively longer time for signal collection (especially of T_1_) and ILT data processing. Therefore, it was decided that to best meet the oil and food industry requirements of a sensor system for rapid on-line evaluation of oil oxidation status it is sufficient to use only data from the LF-NMR T_2_ raw relaxation signals curve rapidly collected from the magnetic field (Figure 6). It should be noted and emphasized that the capacity of the computing system to differentiate between different relaxation curves is significantly higher than human capability. Therefore, using the relatively fast extraction of basic relaxation curves (i.e., a few seconds) may be sufficient to differentiate and classify the oxidation status of the tested oils. Furthermore, to gather more highly relevant information from the LF-NMR and to increase the confidence of the classification of the oils, fast self-diffusion coefficient D data collected by a gradient pulse in the LF-NMR sensor from each tested oil sample was also used. Taking advantage of the previously reported fact that these two collected LF-NMR relaxation parameters were found to be well correlated with conventional lab chemical standard tests of oil oxidation (PV, *p*-AV, and TOTOX) [41], the way for further developing the automation process became simplified.

### 3.2. The Application

Figure 7 summarizes all the elements of the application that includes: a classification engine (Figure 7c) that is a statistical model used to classify the T_2_ raw relaxation curve (Figure 7b) into three different groups (Figure 7d) reflecting the oil sample’s oxidation (Figure 7a).

The results were evaluated and presented in terms of various metrics of accuracy estimated on a testing set, i.e., a set of instances that were not used for training. Figure 8 shows the estimated “accuracy” and “loss” during 30 different training sessions, each one 210 epochs long. In this context, model accuracy is a measure of how well a CNN can classify data inputs into the correct output. Model accuracy is calculated as the proportion of correct outputs. Model loss, on the other hand, is a measure of how well the CNN can fit the training data. It is typically calculated as the difference between the predicted values and the true values of the training data. A low model loss indicates that the CNN can fit the training data well, while a high model loss indicates that the CNN is struggling to fit the data.

In Figure 8, both model accuracy and model loss show that the ability of the CNN to make correct predictions increases over the training epochs. Training was interrupted after 210 epochs. Model loss, on the other hand, is typically used to identify any issues with the model or the training process. Typically, no major issues were found, and the model converged to the desired accuracy.

The convolutional neural network test performances by oxidation classes are presented in Table 2. In this context, accuracy indicates how many times the model was correct as a percentage of correct answers over the total number of attempts. Precision indicates how good the model is at predicting a specific output class [*true positive/(true positive + false positive)*]. Recall indicates how many times the model was able to detect a specific output class [*true positive/(true positive + false negative)*]. High precision means that the model is making very few false positive predictions and is therefore highly accurate in identifying positive instances. High recall means that the model is identifying most of the positive instances and is therefore highly sensitive to the presence of that particular oxidation class. When considering these estimates as a duplet: high precision and high recall indicate a highly accurate model that can detect most of the instances while minimizing the number of false positives; on the other hand, high precision and low recall will be less likely to produce false positives. Combining precision and recall in a single index: The “f-1 score” is another measure of the performance on test data. It is calculated as the harmonic mean of the precision and recall of the model. The F1-score is defined as:F1-score = 2 × (precision × recall)/(precision + recall)

Model performances are summarized in Table 2 where the number of repetitions indicates the number of different networks that were trained independently. At each network reiteration, the term “support” refers to the number of samples in the test set for a particular class. Thus, each model was tested on 390 samples that were not used for training, and 30 different training sessions were performed (on 30 models that were identical in terms of architecture but were initialized randomly and were tested on different testing sets) for a total 11,700 tests, where the total number of tests is the product of repetitions (*n* = 30) times the support size (*n* = 390). The results indicate that the model achieved state-of-the-art performances over a wide range of different samples at different oxidation levels, which resulted in approximately 97% precision for the class ‘Very Bad’, approximately 88% precision for the ‘Fair’ class, and approximately 94% precision for the ‘Good’ class. The rates of false positives and false negatives were low or extremely low, ranging from 1% to 6% depending on the class. Median precision over the entire set was 93% [IQR 87%, 96%]; median recall was 96% [IQR 83%, 98%]. The weighted average F1-score was approximately 0.95 [IQR 0.86, 0.96] which is comparable with state-of-the-art computer pattern recognition system performance.

Further CNN fine-tuning is in process; this is realized by exploring less conventional architectures and increasing the number of oil samples taken from the large available database. Furthermore, emphasis is especially given to improving the accuracy on the ‘Fair’ class by refining the labeling criteria for the oils that are on the threshold between two classes of self-diffusion values in the region between non-oxidized and highly oxidized oil samples. Finer labeling may improve the accuracy of classification for the intermediate oil oxidation class (i.e., ‘Fair’). It should also be noted that for very problematic samples it is possible to carry out a deeper and more detailed analysis of the chemical and structural changes of a specific oil sample using the 2D T_1_-T_2_ LF-NMR sensor briefly described in the introduction (Section 1.4).

As already stated by Gonzalez Viejo et al. [7] and Dos Santos et al. [24], there are many potential applications for AI-based predictive models such as in more accurate determinations of drug expiration dates, beverage quality, sensory descriptor microbial spoilage, oil authenticity, and many others.

In summary, nuclear magnetic resonance (NMR) spectroscopy represents an excellent tool to provide desired information related to food quality, structure, and texture production processes, the nutritional health values of the products, and the economy associated with production through the correct administration of ingredients. However, it is often regarded as very expensive and demands relatively long operation time, data interpretation, and sample preparation. The developments in recent years of improved LF-NMR relaxation sensor methodologies and technologies has significantly reduced the cost of the magnetometer sensors and the associated software. Presently, the introduction of AI-based automation technologies may better meet the requirements of the food industry with an emphasis on a shorter operation time, reliability, robustness, and an affordable cost of using a proton LF-NMR relaxation sensor. There is now a higher probability of a large-scale introduction of magnetic sensorial systems described in this paper to the food field.

## 4. Conclusions

A methodology with potential industrial application for the rapid evaluation of edible oil oxidation level is described. The time needed for measuring the oil’s oxidation is an important industrial parameter; the described semi-autonomic AI-based LF-NMR sensor needs about several minutes, as compared to the time of conventional chemical and spectral common methodologies of days to weeks. To achieve this, a large-scale database of oil oxidation analyses accumulated in recent years was organized and set for the efficient data mining of relevant criteria of the oxidation status. A suitable machine learning model processed by CNN gave a deep convolution neural network model for classification grading of edible oil oxidation levels (non-oxidized oil, partially oxidized oil, highly oxidized oil). The model consists of an initial input of rapid T_2_ relaxation signals using a relatively low-cost LF-NMR magnetometer and a fast processing system.

Several convolutional neural networks (CNNs) were trained, benchmarked, and optimized in order to define the most efficient setting for the classification task of T_2_ signals into three classes describing three different oxidation levels. The output of the CNN model is the classification of the oxidation of the tested oil. The ML model prediction accuracy of classifying the edible oil oxidation status was validated and found to be very accurate.

The results of the study describe an efficient semi-autonomic AI LF-NMR sensor for industrial prediction of edible oil oxidation status that may be a highly important and efficient tool for all the decision makers associated with the food chain. Though this study focused mainly on oil oxidation, the capabilities of an AI LF-NMR sensor are not limited to this alone and have potential to be implemented to more complex food products in future studies.

The model presented solves the two main obstacles of requiring additional time to generate relaxation time spectra with inverse Laplace transform and also needing an expert in analyzing the relaxation times generated. With that, the intelligent NMR system meets the main requirements of the oil and food industry for real-time, fast, and simple testing using a relatively inexpensive sensor to be applied to production units, transport, storage, and cooking facilities. Further ML training and optimization and development of all the additional aspects involved for final application are under further development.

## 5. Patents

Wiesman, Z., Campisi-Pinto, S., Osheter, T., Linder, C., Osheter, A., Semi-Autonomic TD NMR Sensor of Food Safety and Quality, US provisional patent Application No. 11-246 registered by BGN 23 March 2022.

## Figures and Tables

**Figure 1 sensors-23-02125-f001:**
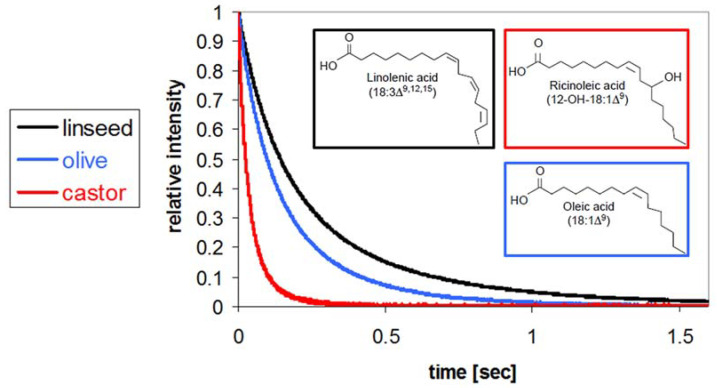
Effect of oils’ chemical composition and structure on low-field nuclear magnetic resonance (LF-NMR) T_2_ relaxation curves. Linseed oil, olive oil, and castor oil with different profiles of unsaturated fatty acids, and therefore different structural organizations, show different rates of proton relaxation curves.

**Figure 2 sensors-23-02125-f002:**
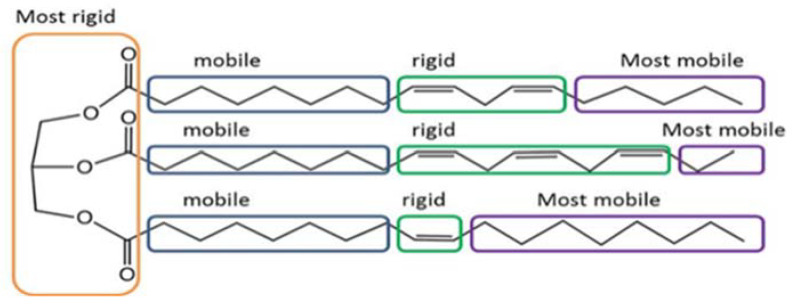
Scheme of triacylglycerol oil structure and segmental motion assigned by segmental rigidity and mobility tests [34].

**Figure 3 sensors-23-02125-f003:**
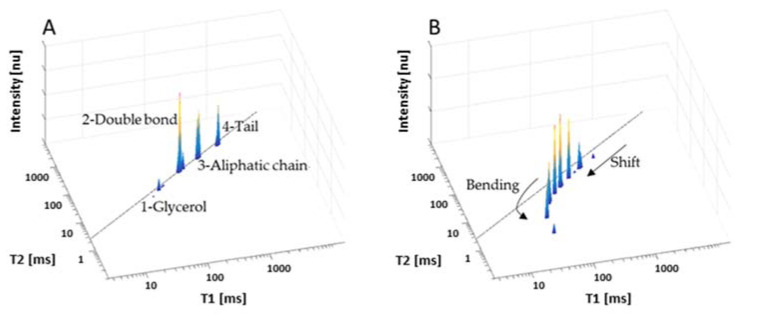
Chemical and morphological time-domain (TD) nuclear magnetic resonance (NMR) sensor 2D T_1_–T_2_ relaxation times of linseed oil before (**A**) and after 120 h of thermal oxidation at 80 °C plus air pumping (**B**). Each peak corresponds to a proton population motion in a different segment of the linseed oil.

**Figure 4 sensors-23-02125-f004:**
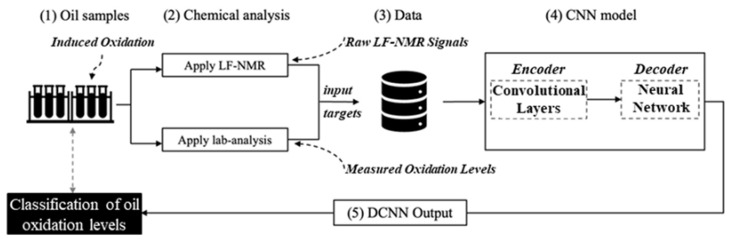
Training and testing system for machine learning: oil samples (1) are analyzed via low-field nuclear magnetic resonance (LF-NMR) and conventional lab methods (2) which are combined into a data frame of inputs and targets (3) for supervised learning via the convolution neural network (CNN) (4); the deep convolution neural network (DCNN) outputs, i.e., classification of T_2_ signals into oxidation classes (5), are benchmarked against ground-truth measurements in order to assess prediction accuracy and to fine-tune the system in a series of recursive cycles until satisfactory accuracy is achieved. Having concluded system fine-tuning and training, the trained CNN is ready for deployment.

**Figure 5 sensors-23-02125-f005:**
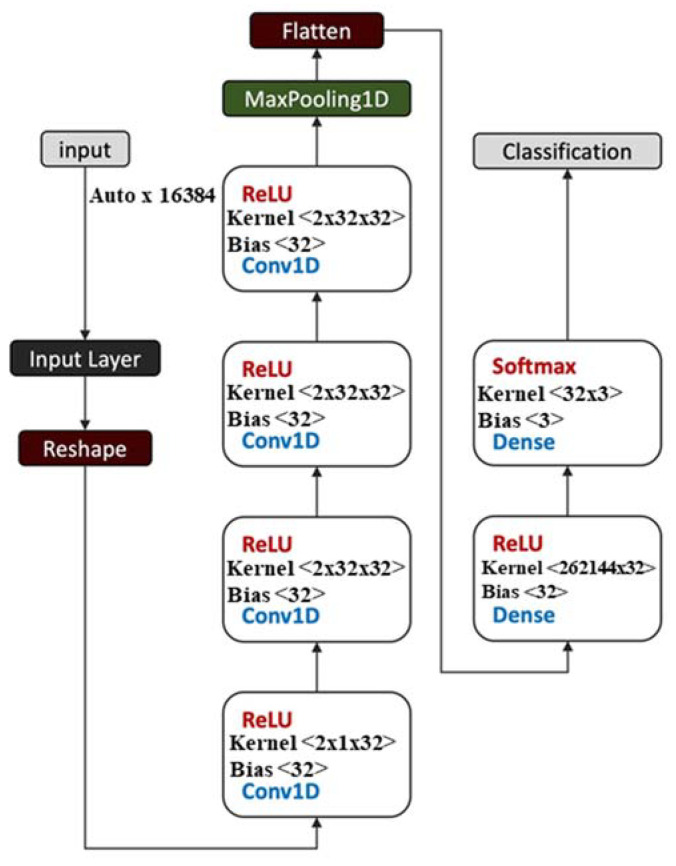
Convolutional neural network (CNN) architecture including input layer, convolution layers, pooling layer, output layer, and classification (Conv1D = Conversion 1D).

**Figure 6 sensors-23-02125-f006:**
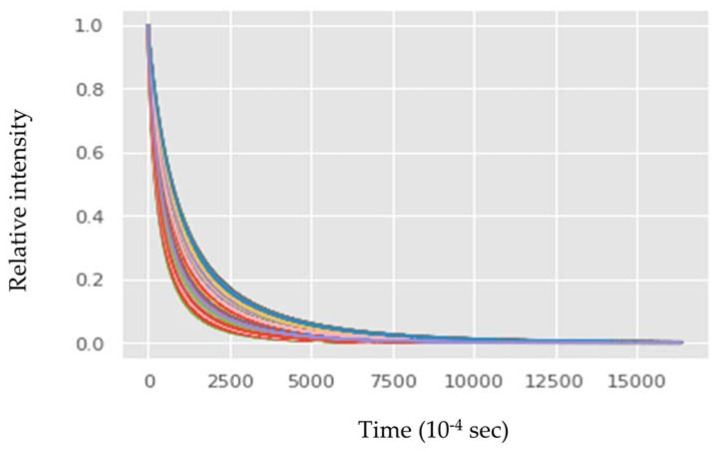
Low-field nuclear magnetic resonance (NMR) T_2_ relaxation time of thermally induced linseed oil sample oxidation. Each line color represents from top to bottom T_2_ relaxation time (0 h, 12 h, 24 h, 48 h, 96 h, and 120 h).

**Figure 7 sensors-23-02125-f007:**
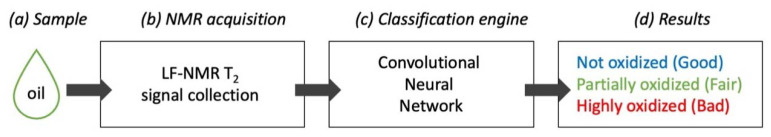
System setup—a typical workflow where a drop of oil is scanned with a low-field nuclear magnetic resonance (LF-NMR) machine; in the next step, the convolutional neural network (CNN) uses the T_2_ signal as an input and returns the oil oxidation class as output.

**Figure 8 sensors-23-02125-f008:**
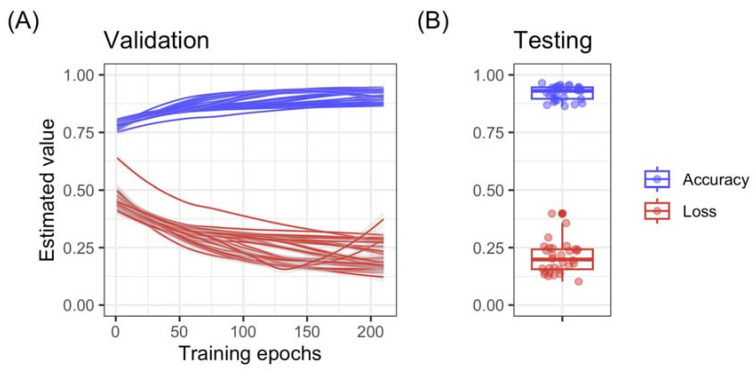
Accuracy and loss functions for 30 different convolutional neural network (CNN) training sessions. (**A**) refers to the validation set; it shows how accuracy and loss evolve over time; typically accuracy increases and loss decreases over epochs. (**B**) shows the final performances on the testing set (a subset of data that was not used for training). Data indicate that both validation and testing performances remain homogeneous over multiple (*n* = 30) randomly initiated training sessions, indicating that the CNN is properly tuned, the architecture is appropriate for the data, and performances are replicable.

**Table 1 sensors-23-02125-t001:** Criteria for dividing oil samples into the three categories ‘Good’, ‘Fair’, and ‘Bad’.

Category	D ^1^ Range (×10^−9^ m^2^/s)	PV ^2^ Range (mmol/kg)	Total Samples
‘Good’	>0.03	<20	126
’Fair’	0.02–0.03	20–50	77
‘Bad’	≤0.02	≥50	187

^1^ D = Diffusion coefficient. ^2^ PV = Peroxide value.

**Table 2 sensors-23-02125-t002:** Convolutional neural network test performances by oxidation class.

	Oxidation Class			
	Bad	Fair	Good	Overall
Number of repetitions (n)	30	30	30	30
Support (n of samples)	126	77	187	390
Total number of tests	3780	2310	5610	11,700
Precision (%) (median, [IQR ^1^])	97% [87%, 0.98%]	88% [84%, 90%]	94% [93%, 96%]	93% [87%, 96%]
Recall (%) (median, [IQR ^1^])	98% [96%, 100%]	77% [59%, 83%]	97% [96%, 98%]	96% [83%, 98%]
F1-score (median, [IQR ^1^])	0.96 [0.91, 0.98]	0.81 [0.69, 0.86]	0.96 [0.95, 0.97]	0.95 [0.86, 0.96]

^1^ IQR = Interquartile range.

## Data Availability

Not applicable.

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
