# Peer review of "Semi-Autonomic AI LF-NMR Sensor for Industrial Prediction of Edible Oil Oxidation Status"

_sensors, 2023, doi:10.3390/s23042125_

Round 1
Reviewer 1 Report
Can be accepted after minor spell check.
Author Response
As suggested by Reviewer 1, the English all over the text was carefully checked and corrected in the revised version of the manuscript.
Reviewer 2 Report
This seems like a very interesting application of low-field NMR! There are however some issues that are needed to be addressed.
I am not convinced that neural networks are needed to make the classifications described.
According to Figure 2 in the authors’ paper in Foods 2021, 10(6), 1385 the T2 relaxation properties can be described with a few components, and my feeling is that a multiexponential fit and some cutoff values would do as well.
The authors therefore need to show that the use of neural networks is superior to less sophisticated methods.
My guess is that the cool thing with the neural networks actually is that you can get a better and more stable resolution of the T2 components than with the inverse Laplace transform.
Furthermore, the paper is very talkative and seems to have the intention to explain a lot of things. Still a figure like the aforementioned one is needed to understand the information content of the measurements. Likewise, I miss details on the optimization of the neural networks, like numbers on how the data was split into training, validation, and test sets. THe English is also of very varying quality and some parts need significant editing.
Author Response
Reviewer 2 raised the comment/question why the CNN is needed and what is the advantage of this approach in comparison to the multi-exponential inverse Laplace transformation described by our group in Foods 2021, 10(6), 1385 - Our response is the there is a significant reduction of time required for analysis (few seconds) using the CNN approach in comparison to many minutes by the ILT approach and this better fit the industrial call for fast and reliable sensor. In additional the ILT fingerprinting solution is adding higher rate of error in comparison to the approach of direct using raw proton relaxation signals collected from the magnetic field. All these explanations are described in the text.
As suggested by reviewer 2, we added in the text deeper explanation and information of the measurements (including CNN architecture, etc). We also added details on the optimization of the neural networks including numbers on how the data was split into training, validation, and statistical tests (including Performances metrics).
As suggested, we also improved the English all over the paper and made a better editing work.
Reviewer 3 Report
This work titled 'Semi-Autonomic AILF-NMR sensor for industrial prediction of edible oil oxidation status' is well designed and conducted with results well interpreted. this work evaluated the oils oxidation status which is important to industrial aspect. I am supportive of publication.
Author Response
We highly appreciate the fact that reviewer 3 is satisfied with the paper as is.
Reviewer 4 Report
In this article, the evaluation of oils oxidation status was mentioned as a highly important industrial aspect related to oil quality, health value, production, transportation, storage, and cooking. The oil and food industry is constantly seeking a real-time non-destructive, fast, robust and low-cost sensor for material characteristics.
Thematically the work is interesting for the researchers and professionals and the proposed manuscript is relevant to the scope of the journal.
I found it appropriate for publication in the Sensors journal, but only after some modifications and clarification from the Authors.
The title is a clear representation of the manuscript's content. The abstract reflects realistically the substance of the work.
The overall organization and structure of the manuscript are appropriate. The paper is well written and the topic is appropriate for the journal.
The aim of the paper is well described and the discussion was well approached, its results and discussion are correlated to the cited literature data.
The literature review is comprehensive and properly done. Perhaps some newer titles could be added to the reference list.
The novelty of the work must be more clearly demonstrated.
The significance of the Work: Given the large number of analyzed data, this is an interesting study with a possible significant impact in this area.
More data concerning the CNN model should be supplied in section 2.7 and in section 3.2.
Statistical interpretation of the analytical data must be more properly presented. The verification of the model should be performed. Model validation is possibly the most important step in the model building sequence.
Other Specific Comments: The work is properly presented in terms of the language. The work presented here is very interesting and well done, it is presented in a compact manner.
The methodology applied in the research is presented in clear manner, so that it is repeatable by other authors.
The results are presented in a logical sequence and the discussion and analysis of the results are properly elaborated.
The main drawback of the paper i s the extent of novelty, or the main novelty in the present work, compared to the works of other researchers? In my opinion, the authors should put additional effort to demonstrate that the present work gives a substantial contribution in the research area.
Author Response
As suggested we carefully went over the references list, corrected some errors and updated it.
As suggested, the novelty of the work was improved to be more clearly demonstrated.
Reviewer 4 asked for more data concerning the CNN model should be supplied in section 2.7 and in section 3.2. - This was carefully and deeply addressed in the revised paper. As suggested more data and explanation including numerical values concerning the CNN model was added (including of CNN architecture, etc.).
Statistical interpretation of the analytical data was improved to be more properly presented.
As suggested by reviewer 4, the issue of model validation was improved and better addressed and presented in the revised version of the paper (including Performances metrics).
As suggested, we invested additional effort to demonstrate that the present work gives a substantial contribution in the research area and better addressed this issue in the revised paper.
Reviewer 5 Report
In this paper, a large oil oxidation database has been established, and an efficient semi autonomous AI LF-NMR sensor has been constructed, which can test and analyze the oxidation level of oils and fats. The proposed artificial intelligence method integrates a large training set, LF-NMR sensor and machine learning, and can be used to develop applications in the oil and food industries. Before considering publishing, there are some questions that need to be clarified. Detailed comments are as follows:
1. It is mentioned that there are reports in recent years that LF-NMR relaxation technology is efficient in fast and accurate monitoring of oil oxidation. What are the differences and advantages between this study and previous ones.
2. How to solve the problems of slow detection rate and single species in traditional methods? What is the principle behind the sensor introduced in this paper that can detect various oil substances?
3. How to control the oxidation level of different oils in the experiment
4. The length of introduction is too long, please adjust the structure of the article, review and simplify this part.
5. It is mentioned in the paper that the sensor can be developed and applied in the petroleum industry. Whether the sensor can maintain high efficiency when encountering complex test environments, such as temperature changes and the presence of organic substances.
6. Please check whether the title number and format in the text are correct.
7. Some tense problems, grammatical problems and the beauty of pictures in the article need to be further improved.
Author Response
Detailed comments of reviewer 5 and our specific responses:
- It is mentioned that there are reports in recent years that LF-NMR relaxation technology is efficient in fast and accurate monitoring of oil oxidation. What are the differences and advantages between this study and previous ones.
Response – Most of the reports dealing with monitoring of oil oxidation are based of conventional colorimetric and common spectral technologies that are limited to only single aspect in each methodology and it takes relatively long time of testing using hazardous chemicals. In the present study, we are using a non-destructive most accurate NMR relaxation technology that is providing information of both the chemical and physical/structural aspects. Using the NMR sensor technology is very fast and the proposed artificial intelligence system based on machine learning model (based on large-scale database of oil oxidation information) is making this system very fast (in seconds) in profiling good-non oxidized and bad-highly oxidized oils. All these aspects are explained in depth in the revised version of the paper.
- How to solve the problems of slow detection rate and single species in traditional methods? What is the principle behind the sensor introduced in this paper that can detect various oil substances?
Response – The problems of slow and tedious detection of oil oxidation in traditional methods are well known and documented, therefore there is a call by the oil and food industry to come up with new detection technologies.
The principle of the NMR sensor for any kind of oil oxidation is that during oil oxidation process there are changes in the chemistry and structural organization of the all that can be rapidly detected in the magnetic field and produce signals that show it in proton relaxation curve. These relaxation curve is different in between non-oxidized oil and oxidized oil sample. Using common methods of machine learning (that is well trained, as described in details in the revised paper) the present artificial intelligence (AI) NMR sensor system is profiling the tested oil and provide answer regarding the class of the tested oil (good or bad) in seconds.
- How to control the oxidation level of different oils in the experiment
Response – As described in the paper methodological part, we stimulated different rate of oil oxidation by heating the oil (thermal oxidation). As stated in the paper, this methodology is well known and established in the literature to induced oxidation.
- The length of introduction is too long, please adjust the structure of the article, review and simplify this part.
Response – As suggested, we significantly improved and simplified the introduction section in the revised paper.
- It is mentioned in the paper that the sensor can be developed and applied in the petroleum industry. Whether the sensor can maintain high efficiency when encountering complex test environments, such as temperature changes and the presence of organic substances.
Response – Indeed the sensor is efficient in a wide range of complex test environments including temperature changes and also presence of organic substances.
- Please check whether the title number and format in the text are correct.
Response – As suggested we checked the title number and format of the text and corrected it in the revised paper.
- Some tense problems, grammatical problems and the beauty of pictures in the article need to be further improved.
Response – As suggested we carefully went over the English and corrected it in the text of the revised paper.
Round 2
Reviewer 2 Report
My suggestion is publish when a figure Figure 2 in the authors’ paper in Foods 2021, 10(6) has been added.
Author Response
As suggested by Reviewer 2, we changed Figure 2 to the version published by our research group in Foods 2021, 10(6), and the reference list was also updated accordingly.
Reviewer 5 Report
accepted
Author Response
We appreciate very much the acceptance of the revised paper by reviewer 5.